# Problematic Social Media Use and Depressive Outcomes among College Students in China: Observational and Experimental Findings

**DOI:** 10.3390/ijerph19094937

**Published:** 2022-04-19

**Authors:** Yonghua Chen, Xi Liu, Dorothy T. Chiu, Ying Li, Baibing Mi, Yue Zhang, Lu Ma, Hong Yan

**Affiliations:** 1School of Public Health, Xi’an Jiaotong University Health Science Center, Xi’an 710000, China; chyh_2006@mail.xjtu.edu.cn (Y.C.); liuxi_214240@xjtufh.edu.cn (X.L.); 2Research Centre on College Students Ideological Education and Practice, Xi’an Jiaotong University, Xi’an 710000, China; 3Community Health Sciences Division, School of Public Health, University of California, Berkeley, CA 94704, USA; dtchiu.79@gmail.com; 4Department of Cardiology, The Second Affiliated Hospital of Xi’an Jiaotong University, Xi’an 710000, China; liying2021a@stu.xjtu.edu.cn; 5Department of Epidemiology and Biostatistic, School of Public Health & Global Health Institute, Xi’an Jiaotong University Health Science Center, Xi’an 710060, China; mibaibing@outlook.com; 6Nursing Department, The Second Affiliated Hospital of Xi’an Jiaotong University, Xi’an 710000, China; zymoon95@126.com; 7Global Health Institute, School of Public Health, Xi’an Jiaotong University Health Science Center, Xi’an 710000, China

**Keywords:** depressive symptoms, social media use, perceived social support, social media violence, loneliness, college students

## Abstract

Aims: Problematic social media use is increasing in China and could be a risk factor for depression. We investigated cross-sectional associations between problematic social media use and depressive outcomes among Chinese college students with potential mediation by perceived social support, social media violence, and loneliness. Thereafter, we evaluated the effectiveness of a one-month group counseling intervention in reducing depressive symptoms related to social media addiction. Methods: Depressive symptoms, social media addiction, perceived social support, social media violence, and loneliness were self-reported among 21,000 college students in Shaanxi province, China. A randomized controlled trial was designed based on the results of the observational study and Satir Transformational Systemic Therapy (STST) among 60 college students assigned to intervention (N = 30) or control/no treatment (N = 30). Self-administered surveys were completed at baseline (T1), at the end of the 1-month intervention (T2), and at 2-month follow-up post-intervention (T3). Results: After controlling for relevant covariates, more problematic social media use was associated with more depressive symptoms (β = 0.18, *p* < 0.001) and depression (OR = 1.08, 95% CI: 1.075, 1.092). Multiple mediation analyses found that perceived social support, social media violence, and loneliness significantly mediated associations between problematic social media use and depressive symptoms (model fit: RMSEA = 0.065, GFI = 0.984, CFI = 0.982). Bootstrapping revealed significant indirect effects of problematic social media use on depressive symptoms through the mediators named above (0.143, 95% CI: 0.133, 0.156). The subsequently informed intervention significantly reduced depressive symptoms at T2 (mean difference: −12.70, 95% CI: −16.64, −8.76, *p* < 0.001) and at T3 (mean difference: −8.70, 95% CI: −12.60, −4.80, *p* < 0.001), as well as levels of social media addiction, perceived social support, social media violence, and loneliness. Conclusions: Problematic social media use is a risk factor for depressive outcomes among Chinese college students, and perceived social support, social media violence, and loneliness mediate this association. STST-based group counseling may reduce depressive symptoms related to high social media usage in this population.

## 1. Introduction

Youth is a critical time for the onset of common mental disorders, especially among those attending college. About 24% of college students from low- and middle-income countries report depressive symptoms [1]. In China, 23.8% of college students had depression by 2020 estimates [2]. This is concerning given depression’s role in serious social and educational impairments [3,4] and its facilitation of suicide. While multiple factors are known to contribute to depression in college students, there is growing interest in the potential influence of problematic social media use on depressive outcomes. Various forms of social media are very popular among college students, including blogs, Wechat, and other online social networks [5,6]. Social media use has become an integral component of connecting with friends and family [7]. As many as 21.4% of college students in China use social media services (e.g., WeChat and QQ), and 40% of the time college students spend on mobile phones is on social media [8]. 

Although most college students’ use of social media is non-problematic, there is a small percentage of users whose usage of social networking sites is excessive or compulsive. Problematic social media use is a behavioral addiction characterized by being overly concerned about social media activity, being driven by uncontrollable urges to log on to or use social media, and devoting time and effort to social media in such great amounts to negative impact on important life areas [9]. Epidemiological studies have adopted validated scales to screen individuals who suffer from social media use [10]. The prevalence of problematic social media use has been found to be rather high in low- and middle-income countries. For example, 15.2% of college students in China have problematic social media use [11], and 19.9% of college students in India that use social media [12]. Problematic social media use has begun to be detrimentally linked to college students’ mental health [9].

Despite the considerable prevalence of problematic social media use, its associations with depressive outcomes have not been well-studied among college students in China. Only one small sample size study from the city of Wuhan, China has showed that problematic social media use was associated with increased risk of depression among college students [13]. More studies are needed to further explore this phenomenon among college students. Much is also unknown about the potential mechanisms by which problematic social media use might increase depressive symptoms in college students. In what little literature exists, however, social media usage has been found to be indirectly associated with depressive symptoms through greater social isolation, lower perceived social support, and loneliness [14]. Moreover, studies have found that social media use was positively associated with social media violence among young people [15], while social media violence was associated with more depressive symptoms among adolescents [16]. Therefore, social media violence may serve as another mediator of the associations between problematic social media use and depressive outcomes among college students [17]. However, the mediating effect of social media violence on associations between problematic social media use and depressive symptoms remains unstudied among college students.

Satir Transformational Systemic Therapy (STST) [18,19] is used to improve relationships and communication within the family structure through focusing on a person’s actions, emotions, and perceptions [20]. This theory was developed by Virginia Satir, who is considered to be one of the primary pioneers of family therapy. The foundational concept of STST is the belief all people are connected through a universal life energy, which can be accessed to achieve transformational change, develop and strengthen relationships, change behaviors, and develop positive life energy [18,19]. According to the STST, the psychological pain people experience is the result of the way they manage their perceptions, expectations, emotions, and behaviors. Therefore, through STST treatment, people can examine their experiences and relationships, develop goals, and work toward change [18,19].

A study showed that STST may increase perceived social support and reduce loneliness of college students [21], as well as ameliorate experiences of social media violence and, therefore, be an appropriate means with which to counteract potential mediators of the social media addiction–depression relation among college students. Thus far, however, no studies (to our knowledge) have attempted to test the effectiveness of STST on reducing problematic social media use to improve depressive symptoms among college students.

To address these knowledge gaps, we conducted two studies. In Study 1, an observational study, we examined cross-sectional associations between problematic social media use and depressive outcomes and the potential mediating effects of perceived social support, loneliness, and social media violence within a large-scale sample of college students in Shaanxi, China (Figure 1). Then, based on the findings of Study 1, we designed a randomized controlled trial in Study 2 to test the effectiveness of an STST-based intervention in college students to improve problematic social media use, perceived social support, loneliness, and social media violence to ultimately reduce depression symptoms.

In particular, we hypothesized that: (1) more problematic social media use was associated with more depressive symptoms and higher prevalence of depression; (2) problematic social media use was associated with lower perceived social support, higher loneliness, and social media violence, which in turn were associated with higher depressive outcomes; (3) the STST-theory-based intervention could reduce problematic social media use, loneliness, and social media violence, and increase perceived social support and ultimately reduce depression symptoms among college students in Shaanxi, China.

## 2. Methods

### 2.1. Observational Study (Study 1)

#### 2.1.1. Study Design and Participants

A cross-sectional study was conducted in 2018 among a total of 21,000 students across Shaanxi Province in China. Stratified multilevel cluster random sampling was used to recruit participants at 30 colleges across Shaanxi, stratified by college type (i.e., colleges directly under the Ministry of Education, as well as local, private, and vocational universities). The detail of sampling is shown in Appendix A. All sampled students were invited to complete a 20 min self-administered Chinese language paper questionnaire in classrooms with support available from trained study research assistants who briefed them on the study and questionnaire. Participants with complete data on social media addiction, social media violence, perceived social support, loneliness, and depressive symptoms were included in the data analyses.

Consent for participation was obtained from the school principals before the survey. It was announced to the students that return of the completed questionnaire implied informed consent by students. Students did not receive incentives for their participation. The study was approved by the ethical committee of Xi’an Jiao Tong University (Registration number 2017-788) on 11 November 2017.

#### 2.1.2. Outcome Variable

Depressive symptoms were assessed by the 20-item Center for Epidemiological Studies Depression Scale (CESD-20) [22], and the scale was translated and validated among Chinese college students [23]. The scale assessed frequency of feelings or behaviors (e.g., feelings of fear, loneliness, fear, and/or failure) during the past seven days across four subscales: depressed affect, somatic complaints, low positive affect, and interpersonal problems. Each item was scored on a 4-point scale from 0 to 3, where 0 = “rarely or none of the time (less than 1 day)” and 3 = “most or all of the time (5–7 days)”. Total scores could range from 0 to 60, with higher score indicating more depressive symptoms. Total CES-D scores were analyzed continuously and also as a binary variable where the conventional cut-off score of 16 or more suggesting probable depression was applied. Previous studies have shown the CES-D to have high reliability in young and middle-aged people (Cronbach’s α = 0.88) [22]. In this study, Cronbach’s α for the CES-D was 0.86.

#### 2.1.3. Exposure and Mediating Variables

Problematic social media use: It was assessed using a modified version of the Facebook Addiction Scale (FAS) [24]. In the questionnaire, we replaced the word “Facebook” with “social media sites”. This scale included eight items and assessed six symptoms related to problematic social media use, including: cognitive and behavioral salience, conflict with other activities, euphoria, relapse and reinstatement, withdrawal, and loss of control [25]. Participants reported on the veracity of statements using a five-point Likert scale ranging from 1 (not true) to 5 (extremely true). Scores were summed such that total scores could range between 8 and 40, with higher scores indicating higher social media addiction. The total score was analyzed continuously. This scale has been used in Chinese college students and showed good reliability [24,25]. Cronbach’s α of the scale was 0.84 in the present study.

Perceived social support: The Chinese version of the Perceived Social Support Scale was used to evaluate the level of perceived social support among students [26]. The scale contains 12 items [27] (e.g., “My friends really try to help me”, “I get the emotional help and support I need from my family”) divided into two dimensions: family support and friend support. Each item utilized a seven-point Likert scale (ranging from 1 = “strongly disagree” to 7 = “strongly agree”). A sum of total score was generated and reflected an individual’s perceived level and degree of social support. Scores were analyzed continuously, and higher scores reflected greater social support. Cronbach’s α for the scale was 0.94 in this study.

Loneliness: Loneliness was assessed by the short-form UCLA Loneliness Scale (ULS-8). The scale included 8 items, such as, “I lack companionship”, “There is no one I can turn to”. Respondents responded using a four-point Likert scale (where 1 = “never” to 4 = “always”), with the summed total score ranging from 8 to 32 [28]. Scores were analyzed continuously, and a higher score indicated a higher degree of loneliness. Cronbach’s α for the scale was 0.77 in the present study.

Social media violence: Social media violence was assessed by the Internet Violence Scale developed by the Chinese University of Hong Kong. Four items were included, and respondents used a four-point Likert scale from 1 = “No” to 4 = “Frequently” to report on items, such as “Experienced trouble from having personal or private information exposed on social networks”, “Experienced deception on social networks”. Scores were analyzed continuously, and higher scores indicated more intense past exposure to social media violence events. The scale has been validated among young adults in China [29]. Cronbach’s α of the scale was 0.68 in the present study.

#### 2.1.4. Covariates

Covariates included age, sex (Male/female), grade (freshman, sophomore, junior, senior, and post-graduate), time spent using social media in the past week (i.e., How much time did you spend on social media in the past week?), primary method used to access social media (i.e., What device do you primarily use to log on to social media sites?), number of people known on social media networks (i.e., How many friends do you have across your social media networks?), smoking (i.e., In the past month, did you smoke?), perceived academic stress (i.e., How do you perceive of your academic stress?), parental relationship satisfaction (How do you satisfy with your relationship with your parents?), and physical exercise time per week (i.e., How much time did you spend on physical activity in the past week?).

#### 2.1.5. Statistical Analysis

Descriptive statistics were calculated for depressive outcomes across socio-demographic characteristics. Chi-square tests (for categorical variables), Kruskal–Wallis test (for continuous variables), and Mann–Whitney U test (for continuous variables) were conducted to test for group differences across covariates.

Multivariable linear and logistic regression models were used to assess the associations of social media addiction, perceived social support, social media violence, and loneliness with depressive symptoms and probable depression (CESD score ≥ 16) adjusting for covariates. Effect sizes were presented as a beta coefficient with a standard error or odds ratio (OR) and 95% confidence interval (CI).

Structural equation modeling was used to gauge the extent of any mediating effects through perceived social support, loneliness, and social media violence between problematic social media use and depressive symptoms. A mediation model was constructed with problematic social media use as the independent variable, perceived social support, loneliness, and social media violence as mediating variables, and depressive symptoms (continuous and binary) as the dependent variable. Goodness of Fit Index (GFI > 0.90), Comparative Fit Index (CFI > 0.90), and Root Mean Square Error of Approximation (RMSEA < 0.08) were used to assess the model fit. The bootstrap procedure was used to test the indirect effect of problematic social media use on depressive symptoms through perceived social support, loneliness, and social media violence. Statistical significance of indirect effects was examined by 95% bias-corrected CI generated after bootstrapping. To adjust for the multiple comparison testing effects, Bonferroni Test was used. An indirect effect was considered statistically significant if the 95% CI did not include zero.

### 2.2. Experimental Study (Study 2)

#### 2.2.1. Intervention Design

Briefly, results from our observational Study 1 showed (1) problematic social media use, loneliness, and social media violence were positively associated and perceived social support was inversely associated with depressive symptoms among college students in Shaanxi, China, and (2) perceived social support, loneliness, and social media violence mediated the associations between problematic social media use and depressive symptoms. Based on principle findings from observational Study 1, we designed an intervention for college students to improve upon depressive symptoms through the associated variables of social media addiction, perceived social support, loneliness, and social media violence using tenets from STST. Students underwent group psychological counseling focusing on interpersonal relationships and network interpersonal communication ability based on self-cognition, aiming to remodel self-cognition, improve ability to perceive social support, build good relationships and reduce loneliness, improve communication skills, and avoid social media violence. Group psychological counseling focused on time management was also conducted, and this counseling aimed to help college students re-recognize their life and value, plan college time, and control online time to avoid problematic social media use (Figure 2). All these interventions were conducted in Chinese. The intervention strategies have been validated in previous studies among college students [30,31].

#### 2.2.2. Participant Recruitment

We conducted a randomized controlled trial (RCT) to test the effectiveness of the designed intervention. College students not involved in observational Study 1 were recruited through messages posted onto several Xi’an Jiao Tong University social media accounts (i.e., their WeChat group). Students were eligible and able to participate if they: (1) were college students that used social media; (2) voluntarily provided informed consent. Eligible students were then randomly assigned to either intervention group (*n* = 30) or control group (*n* = 30) using a random number table until sixty students were enrolled in the trial. Calculations for determining sample size are shown in Appendix A. Chinese written informed consent was provided by the students.

#### 2.2.3. Intervention Process

Prior to the intervention, the intervention and control participants completed a baseline survey to assess depressive symptoms, social media addiction, perceived social support, social media violence, loneliness, and socio-demographic characteristics. Between November and December 2020, the intervention group (*n* = 30) met weekly for a total of five times for group counseling that consisted of two online activities and three offline activities (Appendix A). A private WeChat group was created for intervention group members. The research team issued daily messages to remind the intervention participants through the WeChat group to use the Internet with restraint every day. The message was “Hello everyone, excessive Internet usage can be harmful to your health. How about putting down your mobile phone or go down the network and play a sport, or visit the library and enjoy the feeling of holding and reading a real book, or if you are feeling down, reaching out and chatting with a classmate?”. The control group participated in no organized activities and received no messages. The intervention strategies implemented here had been validated in previous studies among college students [30,31,32] and the intervention was conducted in Chinese.

All intervention and control group participants completed a second survey in January 2021 at the 1-month intervention end and then a third survey in late February 2021 two months after intervention end. The flowchart of the course of intervention is shown in Appendix A.

#### 2.2.4. Statistical Analysis

To evaluate intervention effects, analysis of variance (ANOVA) was applied to further test between-group differences of the outcomes (depressive symptoms and depression) and the mediators (problematic social media use, perceived social support, loneliness, and social media violence), controlling for the baseline levels of outcomes and mediators. Cohen’s *d* effect size was calculated with the formula (M_c_ − M_i_)/SD _pooled_. M_c_ − M_i_ is the difference between the group means at T2 or T3. SD _pooled_ means the pooled standard deviation of mean scores in the combined sample. According to the rule of thumb [33], the Cohen’s *d* effect of 0.2 to 0.3 is defined as a small effect, around 0.5 as a medium effect, and above 0.8 as a large effect. To adjust for the multiple comparison effects, Bonferroni Tests were used. SPSS 24.0 and Analysis of Moment Structure (AMOS) were used for data analysis. Statistical significance was set at *p* < 0.05.

## 3. Results

### 3.1. Observational Study (Study 1)

#### 3.1.1. Depressive Symptoms and Prevalence of Depression among College Students across Socio-Demographic Characteristics

The prevalence of depression was higher among males than females (41.3% vs. 31.6%, *p* < 0.001), and among those with overweight and obesity than those with normal weight (*p* = 0.031). Among undergraduates, the prevalence of depression was higher among sophomore, junior, and senior students than freshman (*p* < 0.001). The prevalence of depression was higher among participants with high/extremely heavy academic stress than those with no/average stress. Participants with ≥ college-educated mothers reported higher prevalence of depression (*p* = 0.029). Similar findings were found for depressive symptoms across sex, grade, academic stress, smoking in past month, primary method used to access social media, and parental relationship satisfaction (Table 1).

#### 3.1.2. Problematic Social Media Use, Perceived Social Support, Loneliness, and Social Media Violence of All Students and by Sex

The average scores were as follows for the scales of social media addiction: 19.72 (±5.15); perceived social support, 62.22 (±13.89); loneliness, 16.27 (±4.19); and social media violence, 5.36 (±2.01) among all college students. The scores of social media violence (males vs. females: 5.82 vs. 5.01, *p* < 0.001) and loneliness (males vs. females: 16.43 vs. 16.16, *p* < 0.001) of males were higher than that of females, while the score of perceived social support of females was higher than that of males (males vs. females: 60.00 vs. 63.87, *p* < 0.001) (Table 2).

The prevalence of the lowest (8) and highest (40) scores on the Social Media Addiction Scale were: the lowest, 3.2%, and the highest, 0.2%, among all college students. Among males, the lowest prevalence was 4.2% and the highest was 0.3%. Among females, the lowest prevalence was 2.4% and the highest was 0.2%.

#### 3.1.3. Adjusted Associations of Social Media Addiction, Perceived Social Support, Loneliness, and Social Media Violence with Depression Symptoms and Probable Depression

After adjusting for all the covariates, we found that more problematic social media use (β = 0.18, *p* < 0.001), lower perceived social support (β = −0.17, *p* < 0.001), higher loneliness (β = 0.36, *p* < 0.001), and higher social media violence (β = 0.14, *p* < 0.001) were positively associated with depression symptoms (Table 3).

When we used probable depression as a binary variable, after adjusting for all the covariates, we found that problematic social media use (OR = 1.083, 95% CI: 1.075, 1.092), perceived social support (OR = 0.967, 95% CI: 0.965, 0.970), loneliness (OR = 1.24, 95% CI: 1.23, 1.25), and social media violence (OR = 1.22, 95% CI: 1.19, 1.25) were associated with probable depression (Table 3).

#### 3.1.4. Structural Equation Modeling Results—Problematic Social Media Use and Depressive Symptoms: The Mediating Effects of Perceived Social Support, Loneliness, and Social Media Violence

Controlling for sex and grade, multiple mediation analyses revealed that perceived social support, social media violence, and loneliness significantly mediated the associations between problematic social media use and depressive symptoms among college students (model fit: RMSEA = 0.065, GFI = 0.984, CFI = 0.982). The total indirect effect of problematic social media use on depressive symptoms was significant, with seven mediational pathways being statistically significant (Figure 3 and Table 4). For example, bootstrapping revealed that problematic social media use was associated with depressive symptoms through perceived social support (indirect effect = 0.013, 95% CI: 0.011, 0.016), and it was serially associated with depressive symptoms through perceived social support and loneliness (indirect effect = 0.01, 95% CI: 0.008, 0.012). The indirect effects of other pathways can be found in Table 4. However, the mediating effects were not significant after considering Bonferroni Tests for multiple comparison effects.

More problematic social media use was associated with lower perceived social support (β = −0.06, *p* < 0.001), more loneliness (β = 0.15, *p* < 0.001), and more social media violence (β = 0.21, *p* < 0.001), which in turn were associated with more depressive symptoms. In addition, loneliness also mediated the associations of perceived social support and social media violence with depression symptoms. Lower perceived social support (β = −0.39, *p* < 0.001) and higher social media violence (β = 0.15, *p* < 0.001) were associated with higher loneliness, which in turn was associated with more depression symptoms (β = 0.39, *p* < 0.001). Lower perceived social support was associated with higher social media violence (β = −0.29, *p* < 0.001), which was associated with higher loneliness (β = 0.15, *p* < 0.001) (Figure 3).

### 3.2. Experimental Study (Study 2)

#### 3.2.1. Demographic Characteristics and Probable Depression of Participants at Baseline

In total, 60 college students participated in the intervention study (50% females), and 55 of them were grade one students. The prevalence of probable depression at baseline in the intervention group (30%) was significantly higher than that in the control group (16.67%) (Table 5).

#### 3.2.2. Between-Group Difference of Depressive Symptoms, Problematic Social Media Use, Perceived Social Support, Social Media Violence, and Loneliness

Depressive symptoms: At baseline (T1), the mean score of depressive symptoms was 16.00 (±10.77) for the intervention group and 13.23 (±10.57) for the control group. The mean difference between intervention and control groups was not significant (2.77, 95% CI: −2.75, 8.28, *p* = 0.319). However, the prevalence of probable depression at baseline in the intervention group was significantly higher than that in the control group (30% vs. 16.67%) (Table 5). At the intervention end (T2), mean depressive symptoms scores were 3.00 in the intervention group versus 15.70 in the control group at T2 (mean difference: −12.70, 95% CI: −16.64, −8.76, *p* < 0.001, Cohen’s *d* = 1.67, large effect size), and at 2 months post-intervention (T3), mean depressive symptom scores were 3.43 in the intervention group versus 12.13 in the control group (mean difference: −8.70, 95% CI: −12.60, −4.80, *p* < 0.001, Cohen’s *d* = 1.15, large effect size) (Table 6). These results indicate that the STST-theory-based group psychological counseling could effectively reduce depressive outcomes among college students in China.

Problematic social media use: At T1, the problematic social media use score was 23.53 (±7.12) for the intervention group and 20.50 (±4.59) for the control group. The mean difference between intervention and control groups was: 3.03, 95% CI: −0.06, 6.13, *p* = 0.055. At T2, the problematic social media use score was lower, at 12.87 in the intervention group versus 21.23 in the control group (mean difference: −8.37, 95% CI: −10.91, −5.83, *p* < 0.001, Cohen’s *d* = 1.70, large effect size), and at T3, the problematic social media use score was 12.67 in the intervention group versus 21.33 in the control group (mean difference: −8.67, 95% CI: −11.45, −5.88, *p* < 0.001, Cohen’s *d* = 1.61, large effect size). The between-group differences were significant at T2 and T3 (Table 6).

Perceived social support: At T1, the score of perceived social support was 67.63 (±12.15) for intervention group and 67.00 (±14.38) for control group. The mean difference between intervention and control groups was: 0.63, 95% CI: −6.25, 7.51, *p* = 0.854. It became 78.50 (±9.05) in the intervention group versus 65.93 (±12.13) in the control group at T2 (mean difference: 12.57, 95% CI: 7.04, 18.10, *p* < 0.001, Cohen’s *d* = −1.17, large effect size), and 79.07 (±9.17) in the intervention group versus 65.50 (±12.71) in the control group at T3 (mean difference: 13.57, 95% CI: 7.72, 19.41, *p* < 0.001, Cohen’s *d* = −1.20, large effect size). The between-group differences were significant at T2 and T3 (Table 6).

**Loneliness:** At T1, the score of loneliness was 18.23 (±3.87) for the intervention group and 17.70 (±3.94) for the control group. The mean difference between intervention and control groups was: 0.53, 95% CI: −1.48, 2.55, *p* = 0.599. It became 13.17 (±3.04) in the intervention group versus 17.07 (±3.52) in the control group at T2 (mean difference: −3.90, 95% CI: −5.60, −2.20, *p* < 0.001, Cohen’s *d* = 1.19, large effect size), and 11.00 (±4.14) in the intervention group versus 15.93 (±4.56) in the control group at T3 (mean difference: −4.93, 95% CI: −7.19, −2.68, *p* < 0.001, Cohen’s *d* = 1.13, large effect size). The between-group differences were significant at T2 and T3 (Table 6).

Social media violence: At T1, social media violence experience scores were 4.87 (±1.14) for the intervention group and 5.17 (±1.76) for the control group. The mean difference between intervention and control groups was: −0.30, 95% CI: −1.07, 0.47, *p* = 0.437. It became 4.30 (±0.84) in the intervention group versus 5.60 (±2.06) in the control group at T2 (mean difference: −1.30, 95% CI: −2.11, −0.49, *p* = 0.002, Cohen’s *d* = 0.83, large effect size), and 4.43 (±0.97) in the intervention group versus 5.87 (±2.32) in the control group at T3 (mean difference: −1.43, 95% CI: −2.35, −0.52, *p* = 0.003, Cohen’s *d* = 0.80, large effect size). The between-group differences were significant at T2 and T3 (Table 6).

The intervention on depressive symptoms, problematic social media use, perceived social support, social media violence, and loneliness remain significant after considering multiple testing effect by Bonferroni Tests (Table 6).

## 4. Discussion

This two-part study is the first of its size to investigate epidemiological associations between problematic social media use and depressive outcomes and then construct and test the effectiveness of an RCT to reduce depressive symptoms related to problematic social media use informed by our observational findings among college students in Shaanxi, China. Our results from the observational study showed more problematic social media use was positively associated with depressive symptoms as well as the risk of probable depression among college students, and such associations were mediated by lower perceived social support, higher loneliness, and experiencing social media violence. From our subsequent RCT of psychological intervention based on STST, we found our intervention was successful in reducing depressive symptoms, problematic social media use, loneliness, and social media violence, as well as increasing perceived social support among college students in China.


*Discussion for Study 1*



*Was problematic social media use associated with depressive outcomes among college students in China?*


Consistent with our hypotheses, we found that problematic social media use was positively associated with depressive symptoms and depression among college students in Shaanxi, China. These findings were consistent with findings of similar studies from other low- and middle-income countries. In one analysis among 384 college students from Afghanistan, problematic social media use was reported to have a positive correlation with depression [34]. Another study of 440 children from India also showed that problematic social media use was a risk factor of depression [35]. However, other studies also reported the positive effects of social media on adolescents, such as connecting with friends, greater levels of self-expression [36], and means of education [5]. Therefore, the effects of social media on depressive symptoms among college students remain not confirmed due to the cross-sectional design of this study and the complicated associations between them. High-quality longitudinal studies with larger sample size in this area are needed.


*Did perceived social support, loneliness, and social media violence mediate the associations between problematic social media use and depressive symptoms among adolescents in China?*


Our mediation analyses provide insights on the mechanisms of the influence of problematic social media use on depressive outcomes. Consistent with our hypotheses, problematic social media use appeared to be able to increase depressive symptoms through reducing perceived social support, increasing loneliness, and social media violence. These findings replicate previous research in other populations demonstrating that more social media use (i.e., physically-isolated interactions) is associated with reduced real-life social support and increased loneliness [37] and social media violence [38]. We proposed that these findings could be due to the nature of communication over social media. Typical interactions on social media consist of simple reactions and written comments, and these interactions are therefore superficial and indirect. Heavy social media usage can therefore serve to create or increase psychological distance between individuals by decreasing face-to-face interactions while simultaneously reducing depth and quality of interactions between family members and friends, thus it has negative impacts on college students’ relationships with family and friends [38,39]. Therefore, heavy use of social media may reduce perceived social support, and indeed, a longitudinal study among 221 college students from the USA found an association between increased problematic use of social media with less tangible social support [40]. This in turn may increase the risk of depressive outcomes.

The mediating role of loneliness between problematic social media use and depressive symptoms was consistent with the results of studies among college students in China [41], which showed that social media users have stronger needs for communication than those who do not use social networks. When they heavily use social media, they may inadvertently replace real-world interactions with virtual relationships, which may then increase perceived social isolation [42]. Heavy users or those with problematic social media use may then feel lonelier and more unsatisfied with interpersonal needs [41], again potentially culminating in depressive symptoms.

Furthermore, we found that problematic social media use could increase social media violence exposure, which in turn increased depressive symptoms among college students. One study found the level of social media usage to be associated with an increased risk of social media violence in a dose–response manner among Canadian adolescents [43]. Given that social media is increasingly becoming omnipresent in the daily routines of most college students, a better understanding of strategies to avoid, manage, or combat social media violence exposure is desirable. Overall, the above findings support interventions increasing perceived social support, reducing social media violence and loneliness to reduce depressive symptoms among college students in China.


*Discussion for Study 2*



*Was STST-theory-based intervention effective in reducing problematic social media use, loneliness, social media violence, and depressive symptoms, and in increasing perceived social support among college students in China?*


Consistent with our hypotheses, STST-based cognitive reconstruction group counseling was effective in improving the ability of interpersonal relationships, network interpersonal communication, and time management among Chinese college students. These psychological interventions could reduce problematic social media use, social media violence and loneliness, and increase perceived social support, thus improved the depressive outcomes of college students. Though the prevalence of probable depression in the intervention group was higher than that of control group at baseline, after one-month intervention, the depressive symptoms of the intervention group were significantly lower than that of the control group. Furthermore, depressive symptom scores in the control group did not change significantly throughout the course of our RCT. Thus, these findings further support the effectiveness of STST-theory-based group psychological counseling in improving the depressive outcomes of Chinese college students. Findings from studies in other Chinese adults also showed that an intervention based on the STST can be effective in ameliorating mental health problems [30,44,45]. However, the small sample size of our intervention study precludes its generalizability to other college students. Future large-scale intervention studies employing STST against social media use are needed to test its effectiveness among college students.

This study has several strengths. First, the study design combined a large-scale cross-sectional observational study and an RCT to examine the effects of decreasing problematic social media use, social media violence, and loneliness, and increasing perceived social support on depressive outcomes among college students. All the scales used in this study were validated in the Chinese population.

### Limitations of This Research

This study also has some limitations. First, the cross-sectional design of Study 1 prevents inference of causality. Second, the survey sampled students in Shaanxi province in China, so findings are not generalizable to students in other provinces or beyond China. However, the sample of Study 1 encompassed all types of college-level schools in China, and all majors and grades in these schools; therefore, our findings could provide useful insights for the design of interventions to address problematic social media use and improve mental health in a broad range of college students. Third, data were self-reported and thus may be subject to response bias. Fourth, we did not inform the college students with probable depression screened by the CESD in Study I. Future studies may inform the students with probable depression and take some strategies to improve their mental health after screening. Fifth, the sample size in our RCT was small, limiting the representativeness of our sample. Large-scale RCTs based on the STST theory and our findings will be necessary to further test the effectiveness of these intervention strategies on improving depressive outcomes of college students. Sixth, we did not consider the potential influence of COVID-19 on the effectiveness of our RCT study, as COVID-19 happened before the conducting of the RCT, and it may be related to depressive outcomes of adolescents. Future studies should measure and consider this factor.

## 5. Conclusions

In the future, longitudinal studies are needed to establish the causal relationships between problematic social media use and depressive outcomes among college students. Large-scale RCTs are also needed to further evaluate the effectiveness of intervention strategies used in this study. Still, our findings have significant implications for decision-makers, as well as educational and psychological professionals. We have increased understanding of the potential negative effects of excessive social media use on depressive outcomes among Chinese college students. Findings from the RCT may serve as a model for colleges and government organizations to design large-scale effective STST-theory-based interventions in college settings.

In conclusion, the prevalence of problematic social media use and depression was high among college students in China. Problematic social media use is a risk factor of depressive outcomes among college students in a certain Chinese province, and perceived social support, loneliness, and social media violence mediate such associations. The STST-based group counseling could reduce the depressive symptoms of some college students in China.

## Figures and Tables

**Figure 1 ijerph-19-04937-f001:**
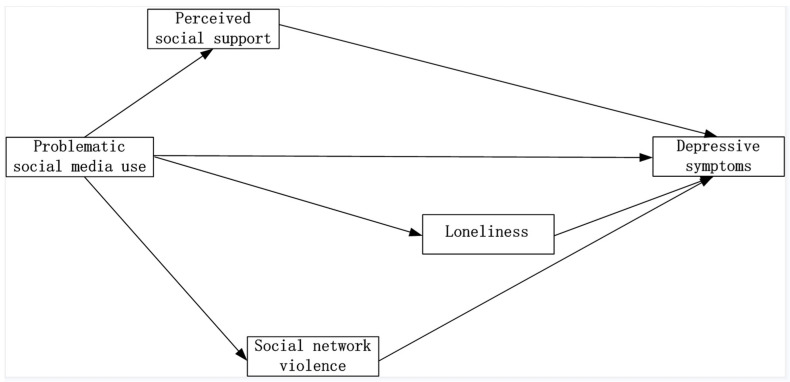
The conceptual model for Study 1.

**Figure 2 ijerph-19-04937-f002:**
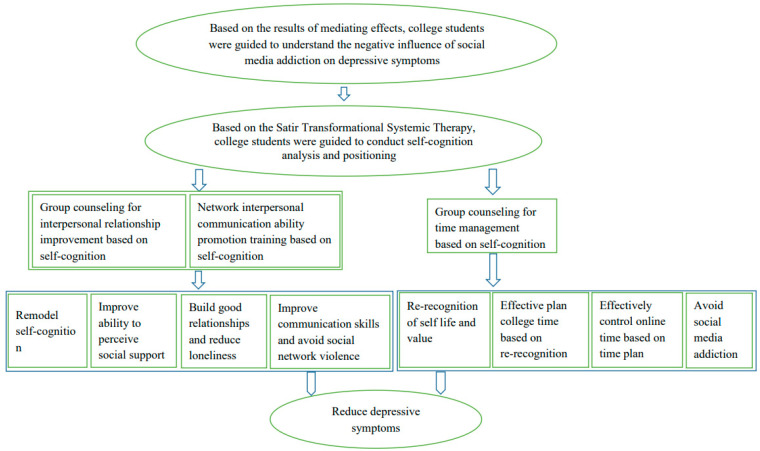
The theoretical model of our psychological intervention based on our findings in Study 1 and the Satir Transformational Systemic Therapy.

**Figure 3 ijerph-19-04937-f003:**
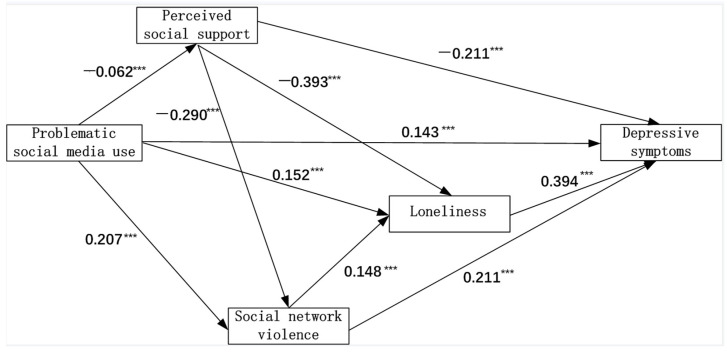
The mediating effects of perceived social support, social media violence, and loneliness on the associations between problematic social media use and depressive symptoms among college students in Shaanxi, China. (Sex and grade were adjusted in this mediation model. RMSEA = 0.065, GFI = 0.984, CFI = 0.982.). ***: *p* < 0.001.

**Table 1 ijerph-19-04937-t001:** Depressive symptoms and prevalence of depression (CESD ≥ 16) among college students in Shaanxi, China by socio-demographic characteristics.

Socio-Demographic Characteristics	*n*	Depressive Symptoms (Mean ± SD)	*p*-Value across Groups ^a^	Depression Prevalence *n* (%)	*p*-Value across Groups ^b^
Sex					
Male	8150	17.6 ± 10.3	**<0.001**	3364 (41.3)	**<0.001**
Female	10,955	15.7 ± 9.4		3466 (31.6)	
Grade					
Freshman	6823	16.0 ± 9.7	**<0.001**	2291 (33.6)	**<0.001**
Sophomore	5489	16.7 ± 9.7		1986 (36.2)	
Junior	3153	17.2 ± 10.0		1235 (39.2)	
Senior	2358	17.3 ± 10.1		954 (40.5)	
Post-graduate	1282	14.6 ± 10.1		364 (28.4)	
Academic stress					
No/relatively low	3458	16.5 ± 10.6	**<0.001**	1248 (36.1)	**<0.001**
Average/general	11,221	15.7 ± 9.5		3667 (32.7)	
Relatively high/extremely heavy	4426	18.5 ± 9.8		1915 (43.3)	
Smoking in past month					
Yes	3243	20.9 ± 10.9	**<0.001**	1757 (54.2)	**<0.001**
No	15,862	15.6 ± 9.4		5073 (32.0)	
Primary method used to access social media					
Computer	1000	19.0 ± 10.8	**<0.001**	466 (46.6)	**<0.001**
Tablet computer	605	24.9 ± 9.9		449 (74.2)	
Smartphone	17,435	16.0 ± 9.6		5871 (33.7)	
Others	65	24.8 ± 11.3		44 (67.7)	
Parental relationship satisfaction					
Dissatisfied	356	23.8 ± 10.6	**<0.001**	227 (63.8)	**<0.001**
A little dissatisfied	2120	20.8 ± 9.9		1132 (53.4)	
Quite satisfied	8925	17.1 ± 9.5		3375 (37.8)	
Very satisfied	7704	14.3 ± 9.5		2096 (27.2)	
Household income					
CNY 0~40,000	11,413	16.4 ± 9.5	0.252	4025 (35.3)	0.335
CNY 40,001~80,000	3962	16.8 ± 10.1		1455 (36.7)	
CNY 80,001~13,000	1983	16.5 ± 10.3		710 (35.8)	
CNY >13,000	1747	16.3 ± 10.6		640 (36.6)	
Highest paternal education					
≤Junior middle school	9750	16.6 ± 9.5	0.106	3479 (35.7)	0.941
Senior middle school/vocational schools	6194	16.4 ± 9.9		2225 (35.9)	
≥College	3161	16.2 ± 10.6		1126 (35.6)	
Highest maternal education					
≤Junior middle school	11,167	16.5 ± 9.5	0.664	3921 (35.1)	**0.029**
Senior middle school/vocational schools	5399	16.4 ± 10.0		1948 (36.1)	
≥College	2539	16.6 ± 10.9		961 (37.8)	

^a^: Mann–Whitney U tests were used to examine the difference of depressive symptoms across sex, smoking in the past month, and parental relationship satisfaction; Kruskal–Wallis tests were used to examine the difference of depressive symptoms across grade, academic stress, primary method used to access social media, household income, highest paternal education, and highest maternal education. ^b^: Chi-square tests were used to examine the difference of prevalence of depression across sex, grade, academic stress, smoking in the past month, primary method used to access social media, parental relationship satisfaction, household income, highest paternal education, and highest maternal education. Numbers in bold indicate statistically significance.

**Table 2 ijerph-19-04937-t002:** Scores for problematic social media use, perceived social support, loneliness, and social media violence for college students in Shaanxi, China: Overall and by sex.

Variable	Sex	Mean ± SD	*t/F*	*p* across Sex
Problematic social media use ^a^	Males	19.64 ± 5.46	−1.88	0.06
	Females	19.78 ± 5.05		
	All	19.72 ± 5.15		
Perceived social support ^b^	Males	60.00 ± 14.57	−19.25	<0.001
	Females	63.87 ± 13.12		
	All	62.22 ± 13.89		
Loneliness ^c^	Males	16.43 ± 4.34	4.37	<0.001
	Females	16.16 ± 4.13		
	All	16.27 ± 4.19		
Social media violence ^d^	Males	5.82 ± 2.26	761.24	<0.001
	Females	5.01 ± 1.76		
	All	5.36 ± 2.01		

^a^: Assessed using the revised version of the Social Media Sites Addiction Scale. ^b^: Assessed by Chinese version of Perceived Social Support Scale. ^c^: Assessed by the short-form UCLA Loneliness Scale (ULS-8). ^d^: Assessed by the Internet Violence Scale developed by the Chinese University of Hong Kong.

**Table 3 ijerph-19-04937-t003:** Adjusted associations between scores on social media addiction, perceived social support, loneliness, and social media violence with depressive outcomes among college students in Shaanxi, China.

Variables			
Model 1: Depressive symptoms (A continuous dependent variable) ^a^	Beta ^b^	SE	*p*
Problematic social media use	0.18	0.01	<0.001
Perceived social support	−0.17	0.01	<0.001
Loneliness	0.36	0.01	<0.001
Social media violence	0.14	0.03	<0.001
Model 2: Probable depression (A binary dependent variable; CESD ≥ 16) ^a^	OR	95% CI	*p*
Problematic social media use	1.083	1.075, 1.092	<0.001
Perceived social support	0.967	0.965, 0.970	<0.001
Loneliness	1.24	1.23, 1.25	<0.001
Social media violence	1.22	1.19, 1.25	<0.001

^a^: Linear regression models were used, the covariates included age, sex, grade, time spent using social media in the past week, primary method used to access social media, number of people known on social media networks, smoking, perceived academic stress, parental relationship satisfaction, and physical exercise time per week. ^b^: Logistic regression models were used, the covariates included age, sex, grade, time spent using social media in the past week, primary method used to access social media, number of people known on social media networks, smoking, perceived academic stress, parental relationship satisfaction, and physical exercise time per week.

**Table 4 ijerph-19-04937-t004:** Mediating effects of perceived social support, loneliness, and social media violence on the associations between problematic social media use and depressive symptoms among college students in Shaanxi, China.

The Paths	Mediating Effect ^a^	95% CI	*p*-Value ^b^	Percentage of Mediating Effects in the Total Effects (%)
1. Problematic social media use → Perceived social support → Depressive symptoms	0.013	0.011, 0.016	0.007	4.545
2. Problematic social media use → Perceived social support → Loneliness → Depressive symptoms	0.01	0.008, 0.012	0.01	3.497
3. Problematic social media use → Perceived social support → Social media violence → Loneliness → Depressive symptoms	0.001	0.001, 0.001	0.008	0.350
4. Problematic social media use → Perceived social support → Social media violence → Depressive symptoms	0.004	0.003, 0.005	0.009	1.399
5. Problematic social media use → Loneliness → Depressive symptoms	0.06	0.055, 0.065	0.013	20.979
6. Problematic social media use → Social media violence → Loneliness → Depressive symptoms	0.012	0.011, 0.014	0.007	4.196
7. Problematic social media use → Social media violence → Depressive symptoms	0.044	0.039, 0.049	0.007	15.385
Indirect effects	0.143	0.133, 0.156	0.003	50.000
Direct effects	0.143	0.134, 0.158	0.011	50.000
Total effects	0.286	0.270, 0.301	0.007	100.000

^a^: Structural equation model was used to analyze the mediating effect of perceived social support, social media violence, and loneliness on the associations between problematic social media use and depressive symptoms. Age and sex were included as covariates. ^b^: We used Bonferroni Tests to adjust for multiple comparison effects, where *p* = 0.007 = 0.05/7 indicates the standard of statistical significance for each mediating effect. These results were not significant after considering multiple comparison effects.

**Table 5 ijerph-19-04937-t005:** The demographic characteristics and probable depression of participants at baseline.

Group	Sex		Grade		Major		Probable Depression at Baseline (%) ^a^	*n*
	Males	Females	First Year	Second Year	Medical	Science		
Intervention group	10	20	28	2	30	0	30.00 *	30
Control group	10	20	27	3	28	2	16.67	30

^a^ Chi-square test was used to compare the probable depression at baseline across intervention and control group. *: *p* < 0.05.

**Table 6 ijerph-19-04937-t006:** Between-group difference of scores on depressive symptoms as well as problematic social media use, perceived social support, social media violence, and loneliness among college students in Shaanxi, China (*n* = 60).

Variables	Time Point	Intervention Group (I)	Control Group (C)	Mean Difference (I-C) (95% CI)	*p*-Values of ANOVA ^a,b^	Cohen’s *d* Effect Size ^c^
Depressive symptoms	T1	16.00 ± 10.77	13.23 ± 10.57	2.77 (−2.75, 8.28)	0.319	
	T2	3.00 ± 4.65	15.70 ± 9.73	−12.70 *** (−16.64, −8.76)	<0.001 *	1.67
	T3	3.43 ± 5.14	12.13 ± 9.36	−8.70 *** (−12.60, −4.80)	<0.001 *	1.15
Problematic social media use	T1	23.53 ± 7.12	20.50 ± 4.59	3.03 (−0.06, 6.13)	0.055	
	T2	12.87 ± 4.90	21.23 ± 4.92	−8.37 *** (−10.91, −5.83)	<0.001 *	1.70
	T3	12.67 ± 5.40	21.33 ± 5.37	−8.67 *** (−11.45, −5.88)	<0.001 *	1.61
Perceived social support	T1	67.63 ± 12.15	67.00 ± 14.38	0.63 (−6.25, 7.51)	0.854	
	T2	78.50 ± 9.05	65.93 ± 12.13	12.57 *** (7.04, 18.10)	<0.001 *	−1.17
	T3	79.07 ± 9.71	65.50 ± 12.71	13.57 *** (7.72, 19.41)	<0.001 *	−1.20
Social media violence	T1	4.87 ± 1.14	5.17 ± 1.76	−0.30 (−1.07, 0.47)	0.437	
	T2	4.30 ± 0.84	5.60 ± 2.06	−1.30 ** (−2.11, −0.49)	0.002 *	0.83
	T3	4.43 ± 0.97	5.87 ± 2.32	−1.43 ** (−2.35, −0.52)	0.003	0.80
Loneliness	T1	18.23 ± 3.87	17.70 ± 3.94	0.53 (−1.48, 2.55)	0.599	
	T2	13.17 ± 3.04	17.07 ± 3.52	−3.90 *** (−5.60, −2.20)	<0.001 *	1.19
	T3	11.0 ± 4.14	15.93 ± 4.56	−4.93 *** (−7.19, −2.68)	<0.001 *	1.13

** *p* < 0.01; *** *p* < 0.001. ^a^: We used Bonferroni Tests to adjust for the multiple comparison testing effect, *p* = 0.003 = 0.05/15 indicates the standard of statistical significance for each intervention effect. ^b^: Baseline level of each variable was adjusted for in the ANOVA. ^c^: Cohen’s *d* effect size was calculated with formula of (Mc-Mi)/SDpooled at T2 or T3. Cohen’s Rules of Thumb suggests that *d* values of 0.2, 0.5, and 0.8 represent small, medium, and large effect sizes, respectively. *: To indicate the significance after considering multiple comparison effect.

## Data Availability

The data that support the findings of this study are available from the corresponding author upon reasonable request.

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
