# Peer review of "Problematic Social Media Use and Depressive Outcomes among College Students in China: Observational and Experimental Findings"

_ijerph, 2022, doi:10.3390/ijerph19094937_

Round 1

Reviewer 1 Report

Abstract:

1. What's the function of STST in this sentence?

A randomized controlled trial was designed based on the results of the observational study and Satir Transformational Systemic Therapy (STST) among 60 college students assigned to intervention (N=30) or control/no treatment (N=30).

Introduction:

1.According to the DSM there is no such thing as a social media addiction. Of course, there is lots of research done and as the researchers clearly describe in their introduction that: 'there is a small number of youth who suffer from their social media use'. It would be much better if this way of framing it is also used in the rest of the introduction. In my opinion, we don't have good scales to 'measure' social media addiction, because we actually don't know how to operationalize it fully (see work of Chris Ferguson or Michelle Colder-Carras). Therefore I wouldn't use the term addiction, but use less firm words. 

2. Strange sentence construction:

The Satir Transformational Systemic Therapy (STST) [14,15], which was used to improve relationships and communication within the family structure through focusing on a person’s actions, emotions, and perceptions

3. The sentence below signals that the depression questionnaire was targeted at social media use specifically, which is not the case I think. I would reframe this sentence.

Thus far, however, no studies (to our knowledge) have attempted to test the effectiveness of STST among college students experiencing depressive symptoms related to high social media usage.

4. Mediators are not discussed in introduction. 

Method:

1. What has been done with youth that scored high on the CES-D (higher than 16), with a probable depression? Have they been informed about that (would be the most ethical thing to do)?

2. Work on social media violence is also mixed (see again work of Chris Ferguson). Also, the cronbach's alpha is rather low, which indicates a not so reliable measure. 

3. Power analysis is missing and that a pity, because 30 participants per condition in an RCT is not enough to find reliable effects. You need at least 50 or more to say something about potential effects of an intervention on depression in this case. Of course, we can't go back in time, but I would like to see this discussed in the discussion and also in the interpretation of effects. 

4. It puzzles me why there was no screening on depressive symptoms in the RCT. If that is the outcome measure, how can you not screen for that? Otherwise changes are high you find floor effects because the participants who were included had no higher levels of depression. 

5. I think that supplemental table 2 should be part of the main paper, and should include more information about the participant sample (not only sex and educational year)

6. This study has been conducted in times of Covid (Nov/Dec 2020), therefore this should be taken into account since this might have had major influences on depression levels, social media usage etc. Could this be integrated as a covariate or was nothing measured regarding Covid? Otherwise, this should at least be mentioned in the discussion. 

7. Analysis strategy RCT is rather simple and could have been more sophisticated, for example a repeated measures ANOVA. Then you make more use of the collected data then with a simple t-test. Furthermore, you cannot control for other factors with simple t-tests. Why was this done? 

Results:

1. Multiple sentences in 3.1.1 signal a within-subject design, which is not the case. E.g.,

Among undergraduates, the prevalence increased with their grade (P<0.001).

2. Has there been a correction for multiple testing? if so, please report that. Otherwise, please do so.

3. What is the function of paragraph 3.1.2?

4. I would like to see effect sizes for the RCT effects.

Discussion:

1. The second part of the claim below cannot be made, this hasn't been tested.

From our subsequent RCT of psychological intervention based on STST, we found our intervention was successful in reducing depressive symptoms among college students in China by reducing social media addiction, social media violence, and loneliness, as well as increasing perceived social support.

2. Please include some sentences on the fact that the first part of the study is purely correlational and therefore we cannot say anything about the direction of effects! At this point the discussion is written very much in the direction of 'social media is bad', but I would like to see a bit more nuanced discussion here. 

3. It might be a better idea to include limitations throughout the discussion, as many of these limitations have an effect on the interpretation of the results, e.g., cross-sectional study and too low n. 

Author Response

Dear reviewer,

The comments have been carefully taken into account and a new revised submission has been uploaded. Please see the attachment.

Reviewer 2 Report

Since technology addiction will be an important social problem in the near future, I find studies on this subject very meaningful. Your study was constructed by making a multi-scale modeling and tested with mediation analysis. Although the title and content of your work are valuable, I tried to write some problems below.

Model Overview

In the model you set up, perceived social support is used as a mediator variable. Whereas, it would be better if perceived social support was the independent variable and social network addiction was the mediating variable. In other words, replacing these two would be more suitable for theory and experience, because perceived social support is an influencing, that is, an independent variable. In other words, the effect of perceived social support on social networks addiction is more suitable for theory and practice. In other words, it is expected that those with high perceived social support will have low addiction. I suggest you rethink this part.

Abstract

Abstract is too long and test results do not need to be written in abstract. Removing the test values ​​will make the abstract easier to read.

Introduction

  1. In the study, the research question indicating which question you are looking for an answer to and hypotheses reflecting the details of the research question were not written. An academic study is incomplete without a research question and hypotheses. These questions should be evaluated in detail by opening sub-headings in the Discussion section.
  2. Your work should be based on a theory. This theory should be explained with at least one paragraph in the Introduction.
  3. The introduction part is mostly written in terms of social media addiction. However, the other variables used in the study are quite insufficient in terms of their relations with each other. I recommend you to improve.
  4. If you can picture the proposed model at the end of the literature and place your hypotheses that you will write on it beforehand, the article will be easy to understand.

Methods

  1. Social media addiction or social media networks? Please decide this. Concepts should be consistent throughout the study.
  2. Why did you use the Facebook Addiction Scale (FAS) even though there are social media addiction measures? You need to explain. What was the reason?
  3. 2 items of each of the measures should be written as an example.
  4. I recommend you to apply the process and approach you wrote in the Experimental study in the observational model.

“Group psychological counseling focused on interpersonal relationships, network interpersonal communication ability based on self-cognition were conducted, aiming to remodel self-cognition, improve ability to perceive social support, build good relationships and reduce loneliness, improve communication skills, and avoid social  media violence”

Discussion

  1. I recommend you to open a subtitle for each Study and evaluate it.
  2. The evaluation of the research question should be in the light of the hypothesis results and current literature under the last sub-title.
  3. Limitations should be written under a separate heading.

Conclusion

  1. A broader introduction should be written. So what were we looking for and what did we find in the end.
  2. Implications for the future should be added.

Reviewer 3 Report

Dear colleagues, 

Thanks to the authors for their efforts in editing the paper. I still have some reservations about several areas of the manuscript. My comments below:

  1. the problem is with your conclusion - please be so kind and add 2-3 paragraphs.
  2. Please implement these articles: about social medias: 

      https://doi.org/10.3390/su131810442  -  Tkacova et. al. 

      https://doi.org/10.3390/su13084138  -   Tkacova et. al. 

      DOI: 10.34291/BV2021/01/Tkacova.

     3. What is a connection between happiness and Social Media Addiction and Depressive  -  https://doi.org/10.3390/su131910826  Petrovic et al. 

     4. Compare your results with: /well - being/ and depresion  https://doi.org/10.15503/jecs2021.1.413.425

and https://doi.org/10.15503/jecs20131.57.70 

  https://doi.org/10.15503/jecs20141.127.144 

sincerely 

Round 2

Reviewer 1 Report

Thank you for taking into account almost all of my previous comments. I still believe it is ethically unsound to not inform people about their mental health status (depression in this case), but that can't be changed at this point unfortunately. Furthermore, I do think that there is a difference in baseline depression scores (The prevalence of probable depression at baseline for the intervention group was 30.00%, and 16.67% for the control group), which might obscure the intervention effects. Can you elaborate about that in your result section? 

Reviewer 2 Report

The study became more understandable in this form.

However, I want to remind you of two things.

But I leave the final decision to you.

  1. In the summary part, giving the values of the analyses in such detail and writing too long will cause the main findings to be overshadowed.
  2. Does perceived social support affect SM addiction or vice versa? I think the first one would be more consistent in real life and logically.
    You say you did it in accordance with the literature, but there are studies in the literature on the contrary.

Reviewer 3 Report

My comments have been incorporated. I recommend the article for publication.

Author Response

Point1:My comments have been incorporated. I recommend the article for publication.

Response: Many thanks!

This manuscript is a resubmission of an earlier submission. The following is a list of the peer review reports and author responses from that submission.